## [Decision Letter · Decision Letter 0]

12 Jan 2026

Dear Dr. Villarreal-Zegarra,

Thank you for submitting your manuscript to PLOS ONE. After careful consideration, we feel that it has merit but does not fully meet PLOS ONE’s publication criteria as it currently stands. Therefore, we invite you to submit a revised version of the manuscript that addresses the points raised during the review process.

Across reviewers, the manuscript is considered clearly written and supported by a well-articulated system architecture. However, significant strengthening is required before further consideration. Reviewer 1 requests the addition of an abbreviations table, clearer articulation of the manuscript structure and contributions, and an updated related work section with a synthesis table. Reviewer 2 highlights the need to address ecological validity (given reliance on simulated interactions), report inter-rater reliability, expand the analysis of cost–effectiveness trade-offs, correct timeline inconsistencies, and broaden the discussion of personas, safety incidents, and future extensions (e.g., fine-tuning or retrieval-augmented generation). Reviewer 3 raises concerns about the novelty, requesting a clearer articulation of the manuscript’s distinct scientific contribution, a more explicit positioning relative to closely related systems, and stronger empirical claims beyond descriptive comparisons.

We look forward to receiving your revised manuscript.

Kind regards,

Issa Atoum

Academic Editor

PLOS One

Journal Requirements:

Reviewers' comments:

Reviewer's Responses to Questions

**Comments to the Author**

1. Is the manuscript technically sound, and do the data support the conclusions?

Reviewer #1: Yes

Reviewer #2: Partly

Reviewer #3: Partly

2. Has the statistical analysis been performed appropriately and rigorously?

Reviewer #1: Yes

Reviewer #2: Yes

Reviewer #3: N/A

3. Have the authors made all data underlying the findings in their manuscript fully available?

Reviewer #1: Yes

Reviewer #2: Yes

Reviewer #3: No

4. Is the manuscript presented in an intelligible fashion and written in standard English?

Reviewer #1: No

Reviewer #2: Yes

Reviewer #3: Yes

Reviewer #1: This paper proposes a cross-sectional study in two phases. developed a mental health platform integrating a conversational agent with functionalities including personalized context, pretrained therapeutic modules, self-assessment tools, and an emergency alert system. Second, evaluated the agent’s responses in simulated interactions based on predefined user personas for each LLM. Four expert raters assessed 816 interaction pairs using a 5-criterion Likert scale evaluating tone, clarity, domain accuracy (correctness), robustness, completeness, boundaries, target language, and safety. In addition, we use performance metrics based on numerical criteria such as cost, response length, and number of tokens. Multiple linear regression models were used to compare LLM performance and assess metric interrelations. First, developed a web-based mental health platform using a user-centered design, structured into frontend, backend, and database layers. The system integrates therapeutic chat (GPT-4o and Llama 3.1-8B), psychological assessments (PHQ-9, GAD89 7), CBT-based tasks, and an emergency alert system. The platform supports secure user authentication, data encryption, multilingual access, and session tracking. Second, GPT4o outperformed Llama 3.1-8B in both performance metrics based on numerical criteria and Likert scale criteria, generating longer and more lexically diverse responses, using more tokens, and scoring higher in clarity, robustness, completeness, boundaries, and target language. However, it incurred higher costs, with no significant differences in tone, accuracy, or safety.

Good work keeps up

But some comments are needed?

submitted to editor

Reviewer #2: This manuscript presents a methodologically rigorous investigation into the development and comparative evaluation of large language model (LLM)-based conversational agents for mental health interventions targeting depression and anxiety. The authors address a critical gap in the literature regarding transparency, reproducibility, and standardization in LLM evaluation for clinical applications. The study is structured in two phases: (1) system design and development documentation, and (2) controlled comparative evaluation of GPT-4o versus Llama 3.1-8B using standardized metrics.

Specific Strengths:

1. Methodological Transparency: The authors provide comprehensive technical documentation of the three-layer system architecture (frontend: React+TypeScript; backend: Express.js; database: PostgreSQL), addressing the critical issue that 99.3% of prior studies evaluated closed-source models without sufficient version identification. This level of transparency significantly enhances reproducibility.

2. Robust Statistical Framework: The sample size calculation is appropriately justified (minimum 788 interaction pairs for d=0.2, α=0.05, power=80%), and the final sample of 816 interaction pairs exceeds this threshold. The use of multiple linear regression models adjusted for language, user persona, and researcher effects demonstrates methodological sophistication.

3. Comprehensive Evaluation Framework: The study employs a dual assessment approach—(a) numerical performance metrics (response length, lexical diversity, token counts, cost analysis) and (b) eight Likert-scale criteria (tone, clarity, domain accuracy, robustness, completeness, boundaries, target language, safety)—which aligns with expert consensus recommendations from prior Delphi studies.

4. Open Science Practices: Full data availability via Figshare (DOI: 10.6084/m9.figshare.29606618.v1) and adherence to CHART reporting guidelines represent commendable scholarly practices.

5. Comparative Inclusion of Open-Source Models: The evaluation of Llama 3.1-8B alongside GPT-4o addresses concerns about over-reliance on proprietary systems and provides valuable insights for resource-constrained implementations.

Major Methodological Concerns:

1. Ecological Validity Limitations: The exclusive reliance on simulated interactions conducted by four research team members, rather than authentic patient-agent exchanges, substantially limits the generalizability of findings to real-world clinical contexts. While the ethical justification for this approach is sound given potential risks from AI hallucinations, the absence of external validation with actual end-users represents a significant limitation.

Recommendation: The authors should explicitly acknowledge this limitation and discuss plans for phased real-world validation studies. Consider proposing a multi-stage validation framework: (a) expert evaluation (completed), (b) pilot testing with supervised participants, (c) randomized controlled trial.

3. Inter-Rater Reliability Omission: Despite employing four independent raters to assess 816 interaction pairs using subjective Likert criteria, the manuscript does not report inter-rater reliability statistics (e.g., intraclass correlation coefficient, Fleiss' kappa). This omission is critical because subjective assessments of constructs like "tone," "clarity," and "robustness" require demonstrated consistency across evaluators to establish measurement validity.

Recommendation: Calculate and report inter-rater reliability coefficients for each Likert criterion. If reliability is suboptimal for certain dimensions, discuss implications for interpretation and propose refinements to assessment protocols.

4. Cost-Effectiveness Analysis Depth: While the study reports that GPT-4o incurs approximately 140-fold higher costs than Llama 3.1-8B ($45.60 vs. $0.32 per 100K output tokens), there is insufficient discussion of the cost-performance trade-offs for implementation decisions. The clinical significance of GPT-4o's superior performance on clarity, robustness, completeness, boundaries, and target language metrics must be weighed against economic constraints in real-world healthcare settings.

Recommendation: Provide a dedicated subsection discussing cost-effectiveness from multiple stakeholder perspectives (public health systems, private clinics, low-resource settings). Consider threshold analyses: at what performance differential would the cost premium be justified?

Minor Technical and Presentation Issues:

1. Temporal Inconsistencies (Lines 154-162): The Methods section states that the protocol received ethical approval on September 19, 2025, with data collection occurring September 20-21, 2025. However, Line 154-155 indicates GPT-4o was selected as "the most widely used OpenAI version at the time of the study (May 2025)". This five-month discrepancy requires clarification. Please verify and correct all study timeline references for internal consistency.

2. Absence of Fine-Tuning and RAG: The authors explicitly note that neither fine-tuning nor Retrieval-Augmented Generation (RAG) was applied to either LLM (Lines 157-158). While this decision ensures comparability of base models, the Discussion should address how these techniques might alter comparative performance and whether future iterations will explore these enhancements.

3. User Persona Documentation: While Supplementary Material 3 contains user persona specifications, the main manuscript provides minimal detail about persona construction methodology, clinical diversity, or alignment with actual patient profiles. Greater transparency regarding persona development would enhance reproducibility and allow readers to assess external validity.

4. Safety Incident Reporting: Although "safety" constitutes one of eight evaluation criteria (defined as absence of stigmatization, guilt induction, inappropriate medication recommendations, or self-harm encouragement), the Results section does not explicitly report whether any unsafe responses were identified. Table 3 shows no significant difference in safety scores between models (Llama 3.1-8B: 4.76±0.51 vs. GPT-4o: 4.77±0.46, p=0.844), but qualitative examples of edge cases or concerning responses would provide valuable context.

5. Limited Model Scope: The evaluation encompasses only two LLMs. While the inclusion of both proprietary and open-source models is valuable, the Discussion should acknowledge emerging alternatives (e.g., Claude 3, Gemini 2.0, DeepSeek, Mistral) and outline criteria for future comparative studies.

Strongly recommended revisions:

4. Add cost-effectiveness analysis subsection with practical implementation guidance

5. Provide qualitative examples of response differences between models

6. Discuss potential impact of fine-tuning and RAG on findings

7. Enhance user persona methodology description

Optional enhancements:

8. Include sensitivity analyses examining whether results vary by user persona or language

9. Propose concrete validation roadmap for progression to real-world testing

Conclusion:

This manuscript makes substantive contributions to the methodological standardization of LLM evaluation in mental health applications. The technical development is sound, the statistical analyses are appropriate, and the comparative framework is valuable for the field. The identified concerns are addressable through targeted revisions that will strengthen the manuscript's impact and clinical applicability. With minor revisions, this work will constitute a significant addition to the digital mental health literature.

Recommendation: Minor Revision

Reviewer #3: Dear Editor and Authors,

Summary of the Study

The study evaluates the design, safety framework, and performance of a large language model (LLM)-based conversational agent intended to support individuals with depressive and anxiety symptoms. The manuscript primarily focuses on system-level integration and a comparative evaluation of two existing language models (GPT-4o and Llama 3.1-8B) using simulated conversational scenarios assessed by expert reviewers. The evaluation relies on predefined quantitative metrics and Likert-scale-based qualitative judgments. While the topic is timely and relevant, the study does not clearly demonstrate a novel system development or a distinct scientific contribution beyond descriptive comparison.

Strengths of the Study

• The manuscript addresses a relevant and contemporary topic in digital mental health and AI-assisted interventions.

• The system architecture and safety considerations are described in a clear and organized manner.

• The authors employ a structured evaluation framework and follow established reporting guidelines.

• The comparative analysis between a proprietary and an open-source LLM is practically informative.

Weaknesses of the Study

• The manuscript does not clearly articulate a novel scientific or methodological contribution.

• The work relies exclusively on existing language models without introducing new algorithms, learning strategies, or adaptive mechanisms.

• Evaluation is limited to simulated interactions and expert ratings, with no real user involvement.

• The study reads largely as a system description and comparison rather than hypothesis-driven research.

Discussion of Specific Areas for Improvement

Major Issues

1. The manuscript does not explicitly define its unique contribution to the literature. It is unclear whether the novelty lies in system design, evaluation methodology, or safety implementation, as all major components rely on established practices.

2. Despite claims of system development, the study does not introduce a new model, algorithm, or adaptive framework. The work is primarily an integration and comparison of pre-existing LLMs.

3. The exclusive use of simulated conversations and expert evaluations limits the ecological validity and weakens the empirical strength of the conclusions.

4. The manuscript lacks a focused comparison with closely related studies to clarify how this work advances beyond previously published systems and evaluations.

Minor Issues

1. Some tables and figures could be reduced or moved to supplementary materials to improve readability.

2. Minor grammatical and stylistic issues are present, particularly in the Discussion section.

3. The manuscript does not report inter-rater reliability metrics for expert assessments.

4. The cost analysis lacks sensitivity analysis and contextual benchmarking.

**Do you want your identity to be public for this peer review?** For information about this choice, including consent withdrawal, please see our Privacy Policy

Reviewer #1: No

Reviewer #2: No

Reviewer #3: **Yes:** ِAli Abbas Abbod

---

## [Author Response · Author response to Decision Letter 1]

3 Feb 2026

PONE-D-25-55991

Development, System Design, Safety, and Performance Metrics of a Conversational Agent for Reducing Depressive and Anxious Symptoms Based on a Large Language Model: The MHAI Study

Across reviewers, the manuscript is considered clearly written and supported by a well-articulated system architecture. However, significant strengthening is required before further consideration. Reviewer 1 requests the addition of an abbreviations table, clearer articulation of the manuscript structure and contributions, and an updated related work section with a synthesis table. Reviewer 2 highlights the need to address ecological validity (given reliance on simulated interactions), report inter-rater reliability, expand the analysis of cost–effectiveness trade-offs, correct timeline inconsistencies, and broaden the discussion of personas, safety incidents, and future extensions (e.g., fine-tuning or retrieval-augmented generation). Reviewer 3 raises concerns about the novelty, requesting a clearer articulation of the manuscript’s distinct scientific contribution, a more explicit positioning relative to closely related systems, and stronger empirical claims beyond descriptive comparisons.

Reply: Thank you very much for your review of the manuscript and for your constructive comments. We address each point below.

Comments to the Author

Reviewer #1:

This paper proposes a cross-sectional study in two phases. developed a mental health platform integrating a conversational agent with functionalities including personalized context, pretrained therapeutic modules, self-assessment tools, and an emergency alert system. Second, evaluated the agent’s responses in simulated interactions based on predefined user personas for each LLM. Four expert raters assessed 816 interaction pairs using a 5-criterion Likert scale evaluating tone, clarity, domain accuracy (correctness), robustness, completeness, boundaries, target language, and safety. In addition, we use performance metrics based on numerical criteria such as cost, response length, and number of tokens. Multiple linear regression models were used to compare LLM performance and assess metric interrelations.

First, developed a web-based mental health platform using a user-centered design, structured into frontend, backend, and database layers. The system integrates therapeutic chat (GPT-4o and Llama 3.1-8B), psychological assessments (PHQ-9, GAD89 7), CBT-based tasks, and an emergency alert system. The platform supports secure user authentication, data encryption, multilingual access, and session tracking.

Second, GPT4o outperformed Llama 3.1-8B in both performance metrics based on numerical criteria and Likert scale criteria, generating longer and more lexically diverse responses, using more tokens, and scoring higher in clarity, robustness, completeness, boundaries, and target language. However, it incurred higher costs, with no significant differences in tone, accuracy, or safety.

Good work keeps up

But some comments are needed?

submitted to editor

Reply: Thank you very much. We believe that the reviewer has a thorough understanding of the study.

Below, we respond to each of the comments.

Reviewer #2:

This manuscript presents a methodologically rigorous investigation into the development and comparative evaluation of large language model (LLM)-based conversational agents for mental health interventions targeting depression and anxiety. The authors address a critical gap in the literature regarding transparency, reproducibility, and standardization in LLM evaluation for clinical applications. The study is structured in two phases: (1) system design and development documentation, and (2) controlled comparative evaluation of GPT-4o versus Llama 3.1-8B using standardized metrics.

Specific Strengths:

1. Methodological Transparency: The authors provide comprehensive technical documentation of the three-layer system architecture (frontend: React+TypeScript; backend: Express.js; database: PostgreSQL), addressing the critical issue that 99.3% of prior studies evaluated closed-source models without sufficient version identification. This level of transparency significantly enhances reproducibility.

2. Robust Statistical Framework: The sample size calculation is appropriately justified (minimum 788 interaction pairs for d=0.2, α=0.05, power=80%), and the final sample of 816 interaction pairs exceeds this threshold. The use of multiple linear regression models adjusted for language, user persona, and researcher effects demonstrates methodological sophistication.

3. Comprehensive Evaluation Framework: The study employs a dual assessment approach—(a) numerical performance metrics (response length, lexical diversity, token counts, cost analysis) and (b) eight Likert-scale criteria (tone, clarity, domain accuracy, robustness, completeness, boundaries, target language, safety)—which aligns with expert consensus recommendations from prior Delphi studies.

4. Open Science Practices: Full data availability via Figshare (DOI: 10.6084/m9.figshare.29606618.v1) and adherence to CHART reporting guidelines represent commendable scholarly practices.

5. Comparative Inclusion of Open-Source Models: The evaluation of Llama 3.1-8B alongside GPT-4o addresses concerns about over-reliance on proprietary systems and provides valuable insights for resource-constrained implementations.

Reply: Since all comments in this section focus on the positive aspects of the manuscript, we address them collectively. Thank you very much for your comments. We believe the second reviewer has correctly identified the manuscript's strongest points.

Major Methodological Concerns:

1. Ecological Validity Limitations: The exclusive reliance on simulated interactions conducted by four research team members, rather than authentic patient-agent exchanges, substantially limits the generalizability of findings to real-world clinical contexts. While the ethical justification for this approach is sound given potential risks from AI hallucinations, the absence of external validation with actual end-users represents a significant limitation.

Recommendation: The authors should explicitly acknowledge this limitation and discuss plans for phased real-world validation studies. Consider proposing a multi-stage validation framework: (a) expert evaluation (completed), (b) pilot testing with supervised participants, (c) randomized controlled trial.

Reply: We agree with the reviewer. We have modified the following limitation:

“Second, the evaluation was conducted using research team members simulating user personas rather than real patient interactions. While this approach limits ecological validity, we considered it appropriate for initial safety evaluation to avoid potential ethical risks associated with exposing vulnerable populations to unvalidated systems. Therefore, in later phases of the study, the platform should be validated in real-world settings. For example, this includes studies with real users (patients) in supervised environments, followed by randomized clinical trials.”

3. Inter-Rater Reliability Omission: Despite employing four independent raters to assess 816 interaction pairs using subjective Likert criteria, the manuscript does not report inter-rater reliability statistics (e.g., intraclass correlation coefficient, Fleiss' kappa). This omission is critical because subjective assessments of constructs like "tone," "clarity," and "robustness" require demonstrated consistency across evaluators to establish measurement validity.

Recommendation: Calculate and report inter-rater reliability coefficients for each Likert criterion. If reliability is suboptimal for certain dimensions, discuss implications for interpretation and propose refinements to assessment protocols.

Reply: In our study, the four evaluators did not rate the same interactions. Each researcher (evaluator) assessed a distinct subset of interaction pairs, generated from different conversations for each LLM. Therefore, due to the study design, it is not possible to validly estimate kappa or ICC across the entire dataset. However, to mitigate evaluator-related bias, we used mixed-effects models adjusted for evaluator, thereby preventing the main analysis from being affected by rater severity or leniency, that is, by an evaluator being systematically more stringent or more lenient.

We have added the following paragraph:

“Fourth, because different evaluators rated different interactions, it was not possible to estimate inter-rater reliability coefficients (i.e., kappa or ICC). However, to mitigate evaluator-related bias, we used mixed-effects models adjusted for evaluator, thereby preventing the main analysis from being affected by rater severity or leniency, that is, by an evaluator being systematically more stringent or more lenient.”

4. Cost-Effectiveness Analysis Depth: While the study reports that GPT-4o incurs approximately 140-fold higher costs than Llama 3.1-8B ($45.60 vs. $0.32 per 100K output tokens), there is insufficient discussion of the cost-performance trade-offs for implementation decisions. The clinical significance of GPT-4o's superior performance on clarity, robustness, completeness, boundaries, and target language metrics must be weighed against economic constraints in real-world healthcare settings.

Recommendation: Provide a dedicated subsection discussing cost-effectiveness from multiple stakeholder perspectives (public health systems, private clinics, low-resource settings). Consider threshold analyses: at what performance differential would the cost premium be justified?

Reply: It was not possible to conduct a cost-effectiveness study because clinical outcomes were not available. Instead, we performed a cost sensitivity analysis of a hypothetical care service, focusing only on token-based costs.

We added an additional limitation:

“Sixth, a cost-effectiveness analysis or clinical benchmarking were not possible because, although information on the cost per input and output token was available, there was no information on the clinical effect of the intervention. Therefore, future studies, such as randomized clinical trials, should include cost-effectiveness evaluations of this type of digital mental health intervention.”

In the method’s section, we added:

“We conducted a one-way sensitivity analysis to contextualize the estimated token-based costs under realistic variations in session intensity. Using the mean prompt and completion token counts observed in our evaluation and the per-token prices. We computed the expected LLM execution cost for a standardized course of care comprising 10 sessions of approximately 45 minutes each. Assuming an interaction rate ranging from 1 to 3 model responses per minute (45–135 responses per session).”

In the results section, we added:

“One-way sensitivity analysis of cost

The estimated total cost per patient by 10 sesions of 45 min each one, ranged from $0.22 to $0.66 for GPT-4o, compared with $0.0017 to $0.0053 for Llama 3.1-8B (see Supplementary Material 7). These results indicate that the relative cost difference is robust to plausible changes in message volume, although our estimates capture only variable token costs and exclude infrastructure and implementation overhead.”

In the discussion section, we added:

“Our study performs a sensitivity analysis based on the assumption of a potential mental health care service and considers token-based costs. Although this perspective is limited, as it does not account for implementation, supervision, data management, infrastructure, and other associated costs, it serves as a proxy that allows potential decision-makers and managers to identify a constant cost for packages of 10 sessions and for each individual session, depending on the model. It should be noted that the evidence suggests that, although the initial implementation process of digital mental health interventions is more costly than traditional care, these costs decrease when large volumes of care are considered [25,26].”

Minor Technical and Presentation Issues:

1. Temporal Inconsistencies (Lines 154-162): The Methods section states that the protocol received ethical approval on September 19, 2025, with data collection occurring September 20-21, 2025. However, Line 154-155 indicates GPT-4o was selected as "the most widely used OpenAI version at the time of the study (May 2025)". This five-month discrepancy requires clarification. Please verify and correct all study timeline references for internal consistency.

Reply: Thank you very much for the clarification. We have modified these sections to avoid ambiguity in the temporal sequence of the study. We added:

“The performance of two LLMs was evaluated: GPT-4o, which was selected for being the most widely used OpenAI version at the time of writing the study protocol (May 2025), and Llama 3.1-8B, which was chosen for being an open-access, easy-to-implement, and free LLM.”

“The research protocol was written and planned in May 2025, was approved by the ethics committee on September 19, 2025, and the evaluation and recruitment took place between September 20 and 21, 2025.”

2. Absence of Fine-Tuning and RAG: The authors explicitly note that neither fine-tuning nor Retrieval-Augmented Generation (RAG) was applied to either LLM (Lines 157-158). While this decision ensures comparability of base models, the Discussion should address how these techniques might alter comparative performance and whether future iterations will explore these enhancements.

Reply: We agree with the reviewer’s comment. Not using RAG could be considered a limitation of the study. Therefore, we have added the following paragraph:

“Fifth, our study did not include fine-tuning or retrieval-augmented generation (RAG) at the model design stage. Although it has been shown that fine-tuned models exhibit performance comparable to general-purpose models for different health-related tasks [25], the use of RAG could represent an alternative to improve the model’s metric performance. Therefore, we estimate that subsequent stages of the study will include the use of RAG with transcripts of psychotherapy recordings, coding patient and therapist responses, and incorporating care manuals or clinical practice guidelines to improve the performance of the responses.”

3. User Persona Documentation: While Supplementary Material 3 contains user persona specifications, the main manuscript provides minimal detail about persona construction methodology, clinical diversity, or alignment with actual patient profiles. Greater transparency regarding persona development would enhance reproducibility and allow readers to assess external validity.

Reply: We added:

“Development of user personas and clinical rationale

The construction of user personas focused on the most frequent patient typologies observed in routine telehealth care at a specialized mental health center (Digital Health Centro Médico) and on the selection of clinically prioritized profiles commonly encountered in real-world practice at this center. The user personas were developed based on real cases that served as reference points and were intentionally selected to maximize clinical diversity across diagnoses and symptom severity levels. For each user persona, a set of clinical characteristics was specified, including diagnosis, severity, reason for seeking help, and patient context, as well as sociodemographic characteristics such as age, gender, occupation, and marital status.

To protect privacy and ensure anonymization, the personas were not direct reproductions of identifiable patients. Sociodemographic and contextual details were modified, combined, or generalized so that no persona corresponded to an identifiable individual. Names, dates, and unique events were not retained. This approach preserves clinical realism while minimizing the risk of reidentification. Supplementary Material 3 provides the complete specifications of the user personas.”

4. Safety Incident Reporting: Although "safety" constitutes one of eight evaluation criteria (defined as absence of stigmatization, guilt induction, inappropriate medication recommendations, or self-harm encouragement), the Results section does not explicitly r

---

## [Decision Letter · Decision Letter 1]

15 Feb 2026

Dear Dr. Villarreal-Zegarra,

Thank you for submitting your manuscript to PLOS ONE. After careful consideration, we feel that it has merit but does not fully meet PLOS ONE’s publication criteria as it currently stands. Therefore, we invite you to submit a revised version of the manuscript that addresses the points raised during the review process.

We look forward to receiving your revised manuscript.

Kind regards,

Issa Atoum

Academic Editor

PLOS One

Journal Requirements:

Reviewers' comments:

Reviewer's Responses to Questions

**Comments to the Author**

Reviewer #1: All comments have been addressed

Reviewer #2: (No Response)

2. Is the manuscript technically sound, and do the data support the conclusions?

Reviewer #1: Yes

Reviewer #2: Partly

3. Has the statistical analysis been performed appropriately and rigorously?

Reviewer #1: Yes

Reviewer #2: Yes

4. Have the authors made all data underlying the findings in their manuscript fully available?

Reviewer #1: Yes

Reviewer #2: Yes

5. Is the manuscript presented in an intelligible fashion and written in standard English?

Reviewer #1: Yes

Reviewer #2: Yes

Reviewer #1: This study presents a conversational agent with multiple functionalities and shows that GPT-4o outperforms Llama 3.1-8B in performance, although at a higher cost. This platform could be used in future clinical trials or real-world implementation studies.

Good work keep up

Reviewer #2: The authors have substantially addressed the major concerns raised in the previous review round. The manuscript demonstrates methodological rigor, transparent reporting, and adherence to CHART guidelines. The comparative evaluation of GPT-4o versus Llama 3.1-8B provides valuable insights into LLM performance for mental health applications.

Strengths of the Revision:

1. Comprehensive Response to Reviewers: All three reviewers' comments have been thoroughly addressed with detailed explanations and appropriate manuscript modifications.

2. Enhanced Methodological Transparency: The addition of seven new limitations strengthens the manuscript's scientific integrity, including acknowledgment of ecological validity constraints, inter-rater reliability limitations, absence of cost-effectiveness analysis, and lack of fine-tuning/RAG implementation.

3. Scientific Contribution Clarification: The newly added paragraph in the Introduction clearly articulates the study's unique contribution: a reproducible comparative evaluation framework integrating standardized conversational quality, safety, and performance metrics under controlled conditions.

4. Improved User Persona Documentation: The detailed methodology for persona development (based on real clinical cases with anonymization protocols) significantly enhances reproducibility.

5. Addition of Sensitivity Analysis: The one-way sensitivity analysis for cost estimation (10 sessions × 45 minutes scenarios) provides practical implementation context.

6. Statistical Rigor: The use of mixed-effects models adjusted for evaluator, language, and user persona effectively mitigates potential biases.

Remaining Limitations:

1. Ecological Validity: While the authors appropriately justify the use of simulated interactions for initial safety evaluation, the manuscript would benefit from a more concrete roadmap for real-world validation phases (e.g., timeline for pilot testing with supervised participants).

2. Clinical Effectiveness Gap: The absence of clinical outcome measures limits the ability to perform cost-effectiveness analysis. Consider elaborating on plans to incorporate validated clinical endpoints (e.g., PHQ-9/GAD-7 pre-post changes) in future trials.

3. Generalizability to Other LLMs: The evaluation is limited to GPT-4o and Llama 3.1-8B. While the authors acknowledge this limitation, explicitly discussing selection criteria for future model comparisons would strengthen the framework's applicability.

Minor Recommendations:

- Consider adding a brief statement on platform scalability and infrastructure requirements for real-world deployment

- Clarify whether the emergency alert system has been tested in pilot scenarios

- Discuss potential regulatory considerations (e.g., FDA guidance on clinical decision support software, if applicable to your jurisdiction)

**Do you want your identity to be public for this peer review?** For information about this choice, including consent withdrawal, please see our Privacy Policy

Reviewer #1: No

Reviewer #2: No

---

## [Author Response · Author response to Decision Letter 2]

17 Feb 2026

PONE-D-25-55991R1

Development, System Design, Safety, and Performance Metrics of a Conversational Agent for Reducing Depressive and Anxious Symptoms Based on a Large Language Model: The MHAI Study

Review Comments to the Author

Reviewer #1: This study presents a conversational agent with multiple functionalities and shows that GPT-4o outperforms Llama 3.1-8B in performance, although at a higher cost. This platform could be used in future clinical trials or real-world implementation studies.

Good work keep up

Reply: Thank you very much for your comments.

Reviewer #2: The authors have substantially addressed the major concerns raised in the previous review round. The manuscript demonstrates methodological rigor, transparent reporting, and adherence to CHART guidelines. The comparative evaluation of GPT-4o versus Llama 3.1-8B provides valuable insights into LLM performance for mental health applications.

Strengths of the Revision:

1. Comprehensive Response to Reviewers: All three reviewers' comments have been thoroughly addressed with detailed explanations and appropriate manuscript modifications.

2. Enhanced Methodological Transparency: The addition of seven new limitations strengthens the manuscript's scientific integrity, including acknowledgment of ecological validity constraints, inter-rater reliability limitations, absence of cost-effectiveness analysis, and lack of fine-tuning/RAG implementation.

3. Scientific Contribution Clarification: The newly added paragraph in the Introduction clearly articulates the study's unique contribution: a reproducible comparative evaluation framework integrating standardized conversational quality, safety, and performance metrics under controlled conditions.

4. Improved User Persona Documentation: The detailed methodology for persona development (based on real clinical cases with anonymization protocols) significantly enhances reproducibility.

5. Addition of Sensitivity Analysis: The one-way sensitivity analysis for cost estimation (10 sessions × 45 minutes scenarios) provides practical implementation context.

6. Statistical Rigor: The use of mixed-effects models adjusted for evaluator, language, and user persona effectively mitigates potential biases.

Reply: We address the six comments collectively, as they focus on the strengths of our study. We thank the reviewer for the comments provided in the previous round; they have helped us strengthen the manuscript and improve its scientific rigor.

Remaining Limitations:

1. Ecological Validity: While the authors appropriately justify the use of simulated interactions for initial safety evaluation, the manuscript would benefit from a more concrete roadmap for real-world validation phases (e.g., timeline for pilot testing with supervised participants).

Reply: We have added the following to the subsection “Implications for future studies”:

“Our study outlines a pathway for future research that addresses the limitations identified and ensures a substantial contribution to the field of digital mental health. First, we require evaluations with real users in controlled settings, that is, a preliminary usability assessment with patients and therapists to ensure that the platform is acceptable, usable, and can be adopted as a complement to psychotherapeutic processes. In addition, we should assess additional requirements that may be valuable to users and were not previously considered. Second, to ensure the ecological validity of the findings in real-world contexts, we must conduct a preliminary effectiveness evaluation in small groups, such as pilot clinical trials, or by using non-experimental methods, such as case series without a control group. This would allow us to determine whether the observed results are replicable in routine care settings and to assess preliminary clinical efficacy. Moreover, it would help identify the optimal platform prescription, such as the frequency of use and the number of sessions.”

2. Clinical Effectiveness Gap: The absence of clinical outcome measures limits the ability to perform cost-effectiveness analysis. Consider elaborating on plans to incorporate validated clinical endpoints (e.g., PHQ-9/GAD-7 pre-post changes) in future trials.

Reply: We have added the following to the subsection “Implications for future studies”:

“Second, to ensure the ecological validity of the findings in real-world contexts, we must conduct a preliminary effectiveness evaluation in small groups, such as pilot clinical trials, or by using non-experimental methods, such as case series without a control group. This would allow us to determine whether the observed results are replicable in routine care settings and to assess preliminary clinical efficacy. Moreover, it would help identify the optimal platform prescription, such as the frequency of use and the number of sessions. Third, we should conduct robust clinical studies with control groups and adequate sample sizes. This would allow us to establish clinical efficacy under controlled conditions.”

3. Generalizability to Other LLMs: The evaluation is limited to GPT-4o and Llama 3.1-8B. While the authors acknowledge this limitation, explicitly discussing selection criteria for future model comparisons would strengthen the framework's applicability.

Reply: Added to the limitations subsection:

“First, we evaluated only two LLMs, GPT-4o and Llama 3.1-8B, without including other open-access models such as DeepSeek, larger Llama variants, or other state-of-the-art models such as Gemini 2.5-Pro, Grok-4, or GPT-o3. This restricts the generalizability of our findings and limits comparison with models that may present different performance–cost profiles. Although our results cannot be directly generalized to other LLMs, prior evidence indicates that state-of-the-art reasoning models generally outperform older or non-reasoning models [28]. Therefore, while our findings may not extend to earlier models, it is plausible that more recent state-of-the-art models would demonstrate similar or superior performance.”

Minor Recommendations:

- Consider adding a brief statement on platform scalability and infrastructure requirements for real-world deployment

Reply: We have added the following to the subsection “Implications for future studies”:

“At the architectural level of MHAI, the platform must implement additional servers to support high traffic volumes through horizontal scalability, while maintaining compliance with data security and privacy standards such as HIPAA. Moreover, MHAI should not operate as an isolated platform; instead, it should be integrated into a real clinical care process and interoperable with electronic health record systems such as OpenMRS, Epic, or Cerner.

Regarding new features and functions, several developments are required. First, the emergency button must be tested under real-world conditions with real users, as only patients can determine whether it is helpful in practice. Although the current implementation is functional and operates correctly, it has not yet been tested in real-world contexts due to ethical constraints. Second, functions that allow users to obtain information from a licensed health professional, schedule a new appointment, or be redirected to a real clinical care pathway should be implemented. Third, additional large language models should be implemented to review conversations and extract clinically relevant information that can be validated by a health professional and subsequently entered into the patient’s electronic health record. For example, one model could identify suicide risk in conversations, while another could detect signs and symptoms that can later be validated and documented in the medical record.”

- Clarify whether the emergency alert system has been tested in pilot scenarios

Reply: We have added the following to the subsection “Implications for future studies”:

“Regarding new features and functions, several developments are required. First, the emergency button must be tested under real-world conditions with real users, as only patients can determine whether it is helpful in practice. Although the current implementation is functional and operates correctly, it has not yet been tested in real-world contexts due to ethical constraints.”

- Discuss potential regulatory considerations (e.g., FDA guidance on clinical decision support software, if applicable to your jurisdiction)

Reply: We have added the following to the subsection “Implications for future studies”:

“Fourth, we should perform implementation studies in primary care and hospital settings to evaluate the real-world impact of MHAI in clinical contexts. At the same time, registration with the relevant regulatory authorities, such as the Food and Drug Administration (FDA) in the United States or the Dirección General de Medicamentos, Insumos y Drogas (DIGEMID) in Peru, will be required if the platform is to be considered a medical device for therapeutic use. Prior to this stage, the platform should be regarded as a tool that may improve overall emotional well-being, rather than as a formal therapy.”

---

## [Decision Letter · Decision Letter 2]

27 Feb 2026

Development, System Design, Safety, and Performance Metrics of a Conversational Agent for Reducing Depressive and Anxious Symptoms Based on a Large Language Model: The MHAI Study

PONE-D-25-55991R2

Dear Dr. Villarreal-Zegarra,

We’re pleased to inform you that your manuscript has been judged scientifically suitable for publication and will be formally accepted for publication once it meets all outstanding technical requirements.

Kind regards,

Issa Atoum

Academic Editor

PLOS One

Additional Editor Comments (optional):

Reviewers' comments:

Reviewer's Responses to Questions

**Comments to the Author**

Reviewer #2: All comments have been addressed

2. Is the manuscript technically sound, and do the data support the conclusions?

Reviewer #2: Yes

3. Has the statistical analysis been performed appropriately and rigorously?

Reviewer #2: Yes

4. Have the authors made all data underlying the findings in their manuscript fully available?

Reviewer #2: Yes

5. Is the manuscript presented in an intelligible fashion and written in standard English?

Reviewer #2: Yes

Reviewer #2: The authors have thoroughly addressed all comments raised in the previous rounds of review. The manuscript presents a methodologically rigorous study on the development and comparative evaluation of an LLM-based conversational agent for mental health applications. The study objectives are clearly stated, the statistical analyses are appropriate and well-justified (multiple linear regression adjusted for language, user persona, and evaluator), and the sample size is adequately powered. All data are fully available in a public repository (Figshare), and the study adheres to CHART reporting guidelines. The limitations are transparently acknowledged, including ecological validity constraints, the use of simulated interactions, and the absence of fine-tuning or RAG. The manuscript is clearly written in standard English. I recommend acceptance in its current form.

**Do you want your identity to be public for this peer review?** For information about this choice, including consent withdrawal, please see our Privacy Policy

Reviewer #2: No

---

## [Editor Report · Acceptance letter]

PONE-D-25-55991R2

PLOS One

Dear Dr. Villarreal-Zegarra,

I'm pleased to inform you that your manuscript has been deemed suitable for publication in PLOS One. Congratulations! Your manuscript is now being handed over to our production team.

Kind regards,

on behalf of

Dr. Issa Atoum

Academic Editor

PLOS One